# A Comparison of Methods to Estimate Additive–by–Additive–by–Additive of QTL×QTL×QTL Interaction Effects by Monte Carlo Simulation Studies

**DOI:** 10.3390/ijms241210043

**Published:** 2023-06-12

**Authors:** Adrian Cyplik, Jan Bocianowski

**Affiliations:** Department of Mathematical and Statistical Methods, Poznań University of Life Sciences, Wojska Polskiego 28, 60-637 Poznań, Poland; adrian.cyplik@up.poznan.pl

**Keywords:** biometrical genetics, homozygous lines, Monte Carlo simulation study, phenotypic observations, quantitative trait loci, regression analysis, three-way interaction, weighted regression

## Abstract

The goal of the breeding process is to obtain new genotypes with traits improved over the parental forms. Parameters related to the additive effect of genes as well as their interactions (such as epistasis of gene–by–gene interaction effect and additive–by–additive–by–additive of gene–by–gene–by–gene interaction effect) can influence decisions on the suitability of breeding material for this purpose. Understanding the genetic architecture of complex traits is a major challenge in the post-genomic era, especially for quantitative trait locus (QTL) effects, QTL–by–QTL interactions and QTL–by–QTL–by–QTL interactions. With regards to the comparing methods for estimating additive–by–additive–by–additive of QTL×QTL×QTL interaction effects by Monte Carlo simulation studies, there are no publications in the open literature. The parameter combinations assumed in the presented simulation studies represented 84 different experimental situations. The use of weighted regression may be the preferred method for estimating additive–by–additive–by–additive of QTL–QTL–QTL triples interaction effects, as it provides results closer to the true values of total additive–by–additive–by–additive interaction effects than using unweighted regression. This is also indicated by the obtained values of the determination coefficients of the proposed models.

## 1. Introduction

The goal of breeding is to obtain the most economically advantageous varieties. A very important aspect is to shorten the breeding process without losing the quality of the obtained genotypes. Knowledge of the genetic structure and genes actions can contribute to genetic advances resulting in a shorter breeding process. In traditional quantitative genetics, organisms are studied on the basis of only phenotypic observations, and conclusions about the genotype are drawn based on the ways in which quantitative traits are inherited, among other things. This is usually realized by describing the action of genes using genetic parameters, mean and variance functions of phenotypes [1]. Among these parameters is the additive effect of genes and interactions between them [2]. Additive effects are perpetuated in a population as it increases its homozygosity in subsequent generations. Thus, the significant additive effect of genes in the population means that selection starting in early generations offers hope of obtaining transgenic homozygous lines [3].

The introduction of different types of markers into genetic and breeding research has revolutionized these scientific disciplines. Markers, among other molecular markers, are used in selection and for genetic transformation [4]. Molecular markers show genetic differences between individuals with opposite traits in the form of nucleotide sequences and have been developed and used to speed up the breeding process in many crops [5]. Identifying markers significantly associated with plant traits can help develop new varieties with desirable traits. It is very important to localize the genes or groups of genes controlling a trait (that is, to give the linkage distances of the genes studied to the markers) and determine their effects [6]. The accuracy and correctness of the conclusions depend largely on the precision of determining the location of the gene on the chromosome and the estimation of parameters associated with gene effects [7].

Most important plant traits are quantitative in nature and are influenced by multiple quantitative trait locus (QTLs). The effects of these QTLs on the expression of quantitative trait has been considered in various species such as maize [8], barley [9], rice [10,11], oilseed rape [12,13,14], wheat [15] and sugar beet [16]. Many loci showing cumulative small effects are often rejected in standard QTL analyses [17,18]. However, by interacting with other loci, they can significantly affect the observed quantitative trait. The effect of epistasis (double interaction) on the observed trait has been considered and reported for, among others: wheat [19], *Triticale* [20], tomato [21], soybean [22], rice [23], pea [24], canola [25], maize [26] and barley [27]. However, higher-order interactions are omited, although it is difficult to imagine that a quantitative trait is not determined by the interaction of more than two genes. Based on the knowledge of physiological genetics and biochemistry, interactions between gene products are ubiquitous [28].

A previous report [29] described the results obtained for analytical and numerical comparisons of two methods for estimating the parameter associated with the additive–by–additive–by–additive (*aaa*) interaction effect, namely, the phenotypic method, based on extreme groups of homozygous lines, and the genotypic method, based on marker observations. One of the conclusions of these studies was that the estimate of the total additive–by–additive–by–additive interaction effect based on the marker data was in most cases smaller than that based on the phenotype. The explanation for this phenomenon may be simple. Phenotypic data can be used to estimate only the total three-way interaction effect of all hypothetical gene triplets determining a trait. Using marker data, which can be more or less precisely mapped into the genome, individual gene–by–gene–by–gene interaction effects can be estimated. For practical reasons, the number of QTL–QTL–QTL interactions is deliberately low. The sum of the effects obtained is lower than the phenotypic estimate, and this difference may be reinforced by the lack of markers in the regions where the genes are located.

In addition to the above explanation, other possible sources of differences between the calculated estimates should be considered. The above results concern QTL–QTL–QTL interaction effects obtained by the simplest possible methods, i.e., using multiple linear regression on the marker data. Cyplik et al. [30] showed that modifying this regression by using empirical weights provides a better agreement between phenotypic and genotypic estimates.

The purpose of this paper was to compare two methods for estimating the parameter associated with additive–by–additive–by–additive interaction effects: the phenotypic method, traditionally used in quantitative genetics, and the genotypic method, which is based on marker observation and is now routinely used in many species. The comparison was made using a Monte Carlo simulation study. Consideration was also given to modifying the estimation of the additive–by–additive–by–additive of QTL–QTL–QTL interaction effects by using weighted multiple linear regression (in two variants) to determine a bridge between the two compared methodologies.

In the approach described here, the estimation of the parameter associated with the additive–by–additive–by–additive interaction effect was based on extreme groups of homozygous lines and on data for genotypic markers. Weighted multiple linear regression was used to estimate the QTL–QTL–QTL interaction effects, using standard weighted regression with weights based on estimated variances for lines in two variants: using weighted regression to estimate only QTL–QTL–QTL triples interaction effects (unweighted regression for QTL selection and QTL–QTL epistasis) and using weighted regression on all three stages—for QTL selection, epistatic pairs and QTL–QTL–QTL triples. The effectiveness of the proposed method was tested using Monte Carlo simulations for different experimental variants, such as the number and location of chromosomes of QTL–QTL–QTL triples determining the quantitative trait (Table 1), the presence or absence of QTL–QTL epistasis (Figure 1). Based on the knowledge of the literature, this is the first report of a comparison of methods for estimating a parameter related to the QTL–QTL–QTL triples interaction effect based on simulation studies.

## 2. Results

Difficult experimental situations were examined in this large-scale simulation. Table 2, Table 3, Table 4, Table 5, Table 6, Table 7, Table 8 and Table 9 show the results of the simulation performed to compare the estimates of additive–by–additive–by–additive interaction effects obtained by the four methods: the phenotypic methods and three genotypic methods (one unweighted and two weighted). Table 2, Table 3, Table 4 and Table 5 show the results obtained with an assumed error variance of 5, while Table 6, Table 7, Table 8 and Table 9 show the results when the error variance was 10. Table 2, Table 3, Table 6 and Table 7 contain results assuming that the total effect of epistasis was equal to 0, and Table 4, Table 5, Table 8 and Table 9, assume that the effect of epistasis was significantly different from zero. The results of the simulation studies assuming that all the effects of the *aaa* triples were equal are shown in Table 2, Table 4, Table 6 and Table 8; while Table 3, Table 5, Table 7 and Table 9 contain the results obtained assuming different effects of the *aaa* triples.

### 2.1. Phenotypic Estimates

The results of the simulation showed that the phenotypic estimate was always greater than 15, the true–assumed value, except in four cases for five QTL–QTL–QTL triples: (1) the number of chromosomes containing QTLs was 1, 2, the error variance was equal to 5, the effect of epistasis was different from zero and the effects of *aaa* triples were different (Table 5), (2) the number of chromosomes containing QTLs was 1, 2, the error variance was equal to 10, the effect of epistasis was equal to zero and the effects of all *aaa* triples were equal (Table 6), (3) and (4) the number of chromosomes containing QTLs was 1, 2 (variant code—C10), and 2 (variant code—C11), respectively, the error variance was equal to 10, the effect of epistasis was different from zero, while the effects of *aaa* triples were different (Table 9). The largest values were obtained for one QTL–QTL–QTL triple, also the smallest values for five QTL–QTL–QTL triples (taking into account the analogous distribution of triples in the linkage group). The largest values were obtained for the situation when the QTLs constituting the triple were located in different chromosomes, while the smallest values were for the situation when all three QTLs constituting the triple were located in one linkage group. The phenotypic estimates were larger when assuming no epistasis than when assuming the effect of epistasis on the values of the observed quantitative trait. When considering the effects of triples, equal or different, similar values of phenotypic estimates were obtained. The *aaa_p_* values were larger when the error variance was equal to 10 than for an error variance of 5, except when five triples were assumed.

### 2.2. Genotypic Estimates

The genotypic estimates of *aaa* effects were greater than true–assumed value 15 for 1, 2 and 3 QTL–QTL–QTL triples. For situations where five QTL–QTL–QTL triples were assumed in most cases (58.33%), genotypic estimates were less than 15. With an assumed error variance of 5, in 19 cases (51.78%) genotypic estimates were less than the true–assumed value of *aaa* equal to 15 (Table 6, Table 7, Table 8 and Table 9). The largest values of genotypic estimates (unweighted and weighted) were obtained for one QTL–QTL–QTL triple, also the smallest values for five QTL–QTL–QTL triples. The largest values were obtained for situations when QTLs constituting triples were located in different chromosomes, except for the case when five QTL–QTL–QTL triples with different effects were assumed in the absence of epistasis and an error variance of 10 (Table 8). In contrast, the smallest values for cases when all three QTLs constituting the triple were placed in a single coupling group. In most cases, genotypic estimates were larger for different effects than for equal effects of triples. The genotypic estimates were larger with an assumed larger (10) error variance than with a smaller (5) variance in situations where the quantitative trait was determined by 1, 2 or 3 QTL–QTL–QTL triples. In contrast, when five QTL–QTL–QTL triples were assumed, genotypic estimates were larger for an error variance of 5 than for an error variance of 10.

### 2.3. Phenotypic vs. Genotypic Estimates

The differences between phenotypic estimates and the genotypic estimates for unweighted regression were always positive and were generally smaller when different effects of QTL–QTL–QTL triples were assumed. The relationships ${\bar{\hat{aaa}}}_{p}$ > ${\bar{\hat{aaa}}}_{gw1}$ > ${\bar{\hat{aaa}}}_{gw2}$ > ${\bar{\hat{aaa}}}_{gu}$ were generally observed, except in three cases: (1) when two QTL–QTL–QTL triples were located in three chromosomes for variance error equal to 5, lack of epistasis and different effects of triples (${\bar{\hat{aaa}}}_{gw1}$ = ${\bar{\hat{aaa}}}_{gw2}$ = 17.99, Table 3), (2) when three QTL–QTL–QTL triples were located in two linkage groups, variance error equal to 5, lack of epistasis and different effects of triples (${\bar{\hat{aaa}}}_{gw1}$ > ${\bar{\hat{aaa}}}_{p}$, Table 5), and (3) when three QTL–QTL–QTL triples with equal effects were located in two linkage groups and the effect of epistasis was different from zero: in which case ${\bar{\hat{aaa}}}_{gw1}$ > ${\bar{\hat{aaa}}}_{gw2}$ > ${\bar{\hat{aaa}}}_{p}$ (Table 8). The weighted estimates were always closer to the phenotypic estimates than the unweighted estimates. The weighted method way (i) showed the best (closest to the true assumed value of 15) ratings of *aaa* triples.

### 2.4. Mean Squared Errors

In general, a decrease in the estimates was accompanied by an increase in their mean squared error (Table 2, Table 3, Table 4, Table 5, Table 6, Table 7, Table 8 and Table 9). The relationships of mean squared error for ${\hat{aaa}}_{p}$ < ${\hat{aaa}}_{gw1}$ < ${\hat{aaa}}_{gu}$ < ${\hat{aaa}}_{gw2}$ were generally observed, except when two QTL–QTL–QTL triples with different effects were assumed to be present in two linkage groups in case of lack of epistatic effect and variance error was equal to 5, in which case mean squared error for ${\hat{aaa}}_{gw1}$ > ${\hat{aaa}}_{gw2}$ (Table 3). The mean squared error was larger with an assumed error variance of 10 than with a smaller assumed error variance. The effect values of each triple (equal or different) did not affect the mean squared error values. On the other hand, assuming the presence or absence of the epistasis effect affected the mean squared error values for $\hat{aaa}$: large mean squared error values were obtained when assuming no epistasis. The smallest mean squared errors were obtained when the trait was determined by one QTL–QTL–QTL triple, while the largest was obtained with five QTL–QTL–QTL triples. In general, the relationships of mean squared error values for assumed 1QQQ < 2QQQs < 3QQQs < 5QQQs were observed. The smallest mean squared error was obtained for QTL–QTL–QTL triples located in one linkage group, while the largest was obtained when QTL–QTL–QTL triples were located across three chromosomes.

### 2.5. Coefficients of Determination

Variance explained by the QTL–QTL–QTL triple interactions ranged from 83% (Table 3 and Table 5) to 97% (Table 8). Coefficients of determination for one QTL–QTL–QTL triple assumed ranged from 0.87 to 0.95, with a mean of 0.91. Coefficients of determination for two QTL–QTL–QTL triples assumed ranged from 0.83 to 0.97, with a mean of 0.89. Coefficients of determination for three and five QTL–QTL–QTL triples assumed were similar and ranged from 0.85 and 0.84 to 0.95 and 0.95, with mean of 0.90 and 0.90, for three and five QTL–QTL–QTL triples assumed, respectively.

## 3. Discussion

The goal of the breeding process is to obtain new genotypes with traits improved over the parental forms [31]. Parameters related to the additive effect of genes as well as their interactions (such as additive–by–additive epistasis of gene–by–gene interaction effect and additive–by–additive–by–additive of gene–by–gene–by–gene interaction effect) can influence decisions on the suitability of breeding material for this purpose [32,33,34,35,36,37,38]. Understanding the genetic architecture of complex traits is a major challenge in the post-genomic era, especially for QTL effects, QTL–by–QTL interactions [39] and QTL–by–QTL–by–QTL interactions.

Monte Carlo simulations are a research tool very widely used to solve various problems by obtaining approximate results. Estimation of epistasis effects using simulation studies has been presented, for example, by Bocianowski [40], Ahsan et al. [41], de los Campos et al. [42], Wang et al. [43] and Sharma et al. [44]. With regards to the comparing methods for estimating additive–by–additive–by–additive of QTL×QTL×QTL interaction effects by Monte Carlo simulation studies, there are no publications in the open literature. The paper describes the first simulation study comparing methods for estimating the total additive–by–additive–by–additive effect of gene–by–gene–by–gene interactions. In addition, weighted multiple linear regression was used to estimate QTL–by–QTL–by–QTL interaction effects, with two variants proposed.

The Monte Carlo simulation studies conducted to compare estimation methods could not, of course, take into account all possible experimental situations. It is essential to note that these simulations may not fully capture the complexities and nuances of real-world genetic interactions. The findings should be interpreted with caution until further empirical validation is conducted. However, the parameter combinations used in the analyses presented here correspond to the cases most often encountered in actual QTL studies [45,46,47,48,49,50,51,52,53,54,55,56,57,58,59,60,61,62]. In simulation studies, the number of lines analyzed is most often assumed to be as high as 500 or 2000 [45,46]. Viana et al. [46] analyzed 400 plants, genotyped for 975 SNPs distributed in 10 chromosomes of 100 cm. One thousand SNP markers were considered in their study by Crawford et al. [47]. The ten chromosomes include *Anthoxanthum odoratum*, *Brassica campestris*, *Sorghum sudanense*, *Zea maize*. Practical experience shows that the number of genes located on individual chromosomes varies greatly [48,49,50,51]. The number of QTLs assumed in the present study was 14. The same number of QTLs was presented in their studies by Wu [52] and Balestre et al. [53]. The assumed number of five epistatic pairs was a result observed in many practical experiments [54,55,56,57]. Simulation studies based on 10,000 generations were also presented by Goutelle et al. [58], Avery et al. [59], Johnson et al. [60], Sorojsrisom et al. [61], and Wang et al. [62], among others. The parameter combinations assumed in the presented simulation studies represented 84 different experimental situations. The results obtained from the Monte Carlo simulation studies indicated the stability of the properties of the presented methods of estimating additive–by–additive–by–additive of gene–by–gene–by–gene interaction effect over different types of genetic material. The small effect of error variance on the estimation of additive–by–additive–by–additive of gene–by–gene–by–gene interaction effects by all four methods, and on the conclusions regarding the comparison of the proposed estimation methods, indicates good prospects for the applicability of our conclusions for different plant species.

However, it was found that the number of assumed QTL–QTL–QTL triplets, their assumed positions, in one, two or three chromosomes, and their effects (unequal or equal) affected the estimates and their comparison. Phenotype estimation decreased toward the true value as the number of presumed QTL–QTL–QTL triplets increased (Figure 2). For the largest number of QTL–QTL–QTL triplets, it may have been less than the true value, as genotypic combinations generating extreme lines were not adequately represented in the simulated sample. The phenotypic estimate also tended to the true value when the number of chromosomes with QTL–QTL–QTL triplets decreased, i.e., when there was an accumulation of coupled individual effects. Similarly, phenotypic estimation tended to have a true value when the assumed epistasis effect was different from zero. The phenotypic evaluation was overestimated for a larger assumed error variance. The genotypic estimate decreased its value in a very similar way to the phenotypic estimate. The difference between phenotypic and genotypic estimation was smallest when five QTL–QTL–QTL triplets were assumed, that is when a model close to truly polygenic was used.

The difference between phenotypic and genotypic estimates of the total additive–by–additive–by–additive effect, observed in both numerical [29] and simulation comparisons, is partly due to the fact that genotypic methods do not find all gene–by–gene–by–gene triplets that determine the trait. However, the use of weighted regression shows that the difference between the phenotypic and genotypic estimation can be reduced, and the total genetic effects will be closer to the true value. Similar results were obtained by estimating QTL–QTL–QTL triple interaction effects for total phenolic content in the mapping population of wheat doubled haploid lines under drought stress conditions [30].

As a direct consequence of interactions, particularly the involvement of QTLs in epistatic [63] and higher-order interactions, the effects of single QTL loci are dependent on the genotypes of other loci. As the results presented here show, consideration of the presence of QTL–QTL epistasis effects had an impact on the evaluation of the effect of QTL–QTL–QTL triple interaction. The lack of epistasis assumed in the simulation studies can be considered in two ways: (1) as the actual absence of epistasis effect or (2) as the summed zero effect of QTL–QTL pairs with epistatic effects different from zero. Attempts to use QTLs in breeding programs must consider not only epistatic effects but also higher-order interactions. Determining the contribution of QTL–QTL–QTL triple interactions is important for understanding the genetic basis of complex traits. Therefore, the genetic models for QTL mapping assuming no QTL–QTL–QTL triple interaction can lead to a biased estimation of QTL parameters and QTL–QTL epistasis [63]. This is indicated by the results obtained by Cyplik et al. [64], where genetic parameters were analyzed for selected traits of maize inbred lines. Although the approach presented in this paper is based on, among other things, a genetic map, the proposed methods to estimate additive–by–additive–by-additive of QTL×QTL×QTL interaction effects can be used in association mapping, as a complement and supplement to the analysis of the effects of major genes and epistasis [65,66,67,68].

The total QTLs with additive QTL effects, additive-additive QTL–QTL epistatic effects and additive–additive–additive QTL–QTL–QTL triple interaction effects explained more than 83% of the phenotypic variation. The information obtained in this study will be useful for manipulating the QTLs for plant breeding by marker-assisted selection.

## 4. Materials and Methods

### 4.1. Data Set

Consider a population of *n* significantly differentiated homozygous (recombinant inbred—RI or doubled haploid—DH) plant lines from a cross between two homozygous parents, as is currently practiced in the breeding of selfing species. For this population, an *n*-vector of phenotypic mean observations $y={\left[\begin{array}{cc} \begin{array}{cc} y_{1} & y_{2} \end{array} & \begin{array}{cc} \cdots & y_{n} \end{array} \end{array}\right]}^{\prime }$ and *q n*-vectors of marker genotype observations $m_{l}$, where l=1, 2, ⋯, q, are obtained. The *i*-th element $\left(i=1,\;2,\;\cdots ,\;n\right)$ of vector $m_{l}$ is equal to –1 or 1, depending on the parental genotype exhibited by the *i*-th line.

### 4.2. Estimation of the Additive–by–Additive–by–Additive Interaction Effects of Genes Action

#### 4.2.1. Estimation Based on the Phenotype

The total additive × additive × additive interaction of homozygous loci (three-way interaction) effect on the basis of phenotypic ($aaa_{p}$) observations of total phenolic content can be estimated by the formula [29]:(1)$${\hat{aaa}}_{p}=\frac{1}{2}\left(L_{max}+L_{min}\right)-\bar{L},$$
where $L_{min}$ and $L_{max}$ are the doubled haploid lines with minimal and maximal mean values, respectively; $\bar{L}$ is the mean of all homozygous lines. The number of genes (number of effective factors, $\hat{K}$) obtained, based on phenotypic observations only, was calculated using the formula presented by Kaczmarek et al. [69]:(2)$$\hat{K}=\frac{{\left(L_{max}-L_{min}\right)}^{2}}{4V_{L}},$$
where $V_{L}$ is additive variance.

#### 4.2.2. Estimation Based on the Genotype

A multiple linear regression model [70,71] of trait values on observations of markers acting as explanatory variables [72] was used to map QTL. It allows the elimination of markers that reveal an association with QTL when tested individually, but in multiple analyses are not characterized by any effect on the phenotypic trait [73]. The estimation of *aaa_g_* was based on the assumption that genes responsible for the quantitative trait were completely linked to observed molecular markers. After deciding which *p* markers out of all observed sufficiently well explain the variability of the quantitative trait, we can model phenotypic observation for homozygous lines as
(3)y=1μ+Aβ+Eγ+Tδ+e,
where y—mean values of the observed trait, **1**—the *n*-vector of ones, μ—the general mean, ***A***—*(n × p)*-matrix of the form $A=\left[\begin{array}{cc} \begin{array}{cc} m_{l_{1}} & m_{l_{2}} \end{array} & \begin{array}{cc} \cdots & m_{l_{p}} \end{array} \end{array}\right]$, *l_1_*, *l_2_*, …, *l_p_* ∈ {1, 2, …, *q*}, β—the *p*-vector of unknown parameters of the form $\beta^{\prime }=\left[\begin{array}{cc} \begin{array}{cc} a_{l_{1}} & a_{l_{2}} \end{array} & \begin{array}{cc} \cdots & a_{l_{p}} \end{array} \end{array}\right]$, ***E***—matrix which columns are products of some columns of matrix ***A***, γ—the vector of unknown parameters of the form $\gamma^{\prime }=\left[\begin{array}{cc} \begin{array}{cc} aa_{l_{1}l_{2}} & aa_{l_{1}l_{3}} \end{array} & \begin{array}{cc} \cdots & aa_{l_{p-1}l_{p}} \end{array} \end{array}\right]$, ***T***—matrix which columns are three-way products of some columns of matrix ***A***, δ—the vector of unknown parameters of the form $\delta^{\prime }=\left[\begin{array}{cc} \begin{array}{cc} aaa_{l_{1}l_{2}l_{3}} & aaa_{l_{1}l_{2}l_{4}} \end{array} & \begin{array}{cc} \cdots & aaa_{l_{p-2}l_{p-1}l_{p}} \end{array} \end{array}\right]$, ***e***—the *n*-vector of random variables such that *E*(*e_i_*) = 0, *Cov*(*e_i_, e_j_*) = 0 for *i* ≠ *j*, *i*, *j* = 1, 2, …, *n*. The parameters $a_{l_{1}}$, $a_{l_{2}}$, …, $a_{l_{p}}$ are the additive effects of the genes controlling the quantitative trait, parameters $aa_{l_{1}l_{2}}$, $aa_{l_{1}l_{3}}$, …, $aa_{l_{p-1}l_{p}}$, are the additive×additive interaction effects and parameters $aaa_{l_{1}l_{2}l_{3}}$, $aaa_{l_{1}l_{2}l_{4}}$, …, $aaa_{l_{p-2}l_{p-1}l_{p}}$ are the additive × additive × additive interaction effects. We assume that the epistatic and three-way interaction effects show only loci with significant additive gene action effects. This assumption significantly decreases the number of potentially significant effects and causes the regression model more useful.

##### Unweighted Regression

The selection of markers chosen for model (3) was made by a stepwise trait selection by Akaike Information Criteria [74]. The procedure consisted of two steps: first we divided markers into groups based on the chromosomes they were located on and performed stepwise trait selection by AIC; after that, we combined the remaining markers into one group and repeated selection as above. All of the remaining markers were combined into the final group and the last feature selection was performed on a model with additive × additive × additive interaction effect included. To counteract the multiple comparison problems, we used Bonferroni correction [75].

Denoting by $\alpha^{\prime }=\left[\begin{array}{cc} \begin{array}{cc} \mu & \beta^{\prime } \end{array} & \begin{array}{cc} \gamma^{\prime } & \delta^{\prime } \end{array} \end{array}\right]$ and $G=\left[\begin{array}{cc} \begin{array}{cc} 1 & A \end{array} & \begin{array}{cc} E & T \end{array} \end{array}\right]$ we obtain the model:(4)y=Gα+e.

If ***G*** is of full rank, the estimate of $\alpha_{u}$ from traditional (unweighted) multiple linear regression model is given by [76]:(5)$$\hat{\alpha_{u}}={\left(G^{\prime }G\right)}^{-1}G^{\prime }y.$$

The total three-way interaction *aaa*_gu_ effect of genes influencing the total phenolic content from traditional (unweighted) multiple linear regression model can be found as
(6)$${\hat{aaa}}_{gu}=\sum_{k=1}^{p-2}\sum_{\begin{array}{c} k^{\prime }=2\\ k^{\prime }\neq k \end{array}}^{p-1}\sum_{\begin{array}{c} k^{\prime \prime }=3\\ k^{\prime \prime }\neq k^{\prime } \end{array}}^{p}{\hat{aaa}}_{l_{k}l_{k^{\prime }}l_{k^{\prime \prime }}}.$$

##### Weighted Regression

The modified version of trait regression on marker data in this paper is considered by taking a weighted multiple linear regression, that is, regression with a diagonal matrix ***W*** of unknown variances of observations, which, however, may be empirically found by estimation. In this model, the estimate of $\alpha_{w}$ is:(7)$$\hat{\alpha_{w}}={\left(G^{\prime }W^{-1}G\right)}^{-1}G^{\prime }W^{-1}y,$$
where $W=\left(w_{ii}\right)$ with $w_{ii}$ being the estimated variance for *i*-th homozygous line, *i* = 1, 2, …, *n*. Selection of markers for the weighted regression is made in two ways: (i) by the same method as described for the unweighted case (weighted triple interaction effects only) and (ii) using weighted regression to select markers for model (3) (weighted all effects: QTL, epistatic and triple interaction).

The total three-way interaction {*aaa*_gw1_—for (i) way and *aaa*_gw2_—for (ii) way} effect of genes influencing the quantitative trait from weighted multiple linear regression model can be found as
(8)$${\hat{aaa}}_{gwx}=\sum_{k=1}^{p-2}\sum_{\begin{array}{c} k^{\prime }=2\\ k^{\prime }\neq k \end{array}}^{p-1}\sum_{\begin{array}{c} k^{\prime \prime }=3\\ k^{\prime \prime }\neq k^{\prime } \end{array}}^{p}{\hat{aaa}}_{l_{k}l_{k^{\prime }}l_{k^{\prime \prime }}},$$
where *x* is equal to 1 {for (i) way} or 2 {for (ii) way}.

##### Coefficient of Determination

The coefficients of determination were used to measure how the model (3) fitted the data and, in this study, was the amount of the phenotypic variance explained by total QTLs with additive effects, QTL–pairs with epistatic effects and QTL–triples with additive–by–additive–by–additive effects (*R*^2^). The *R*^2^ was calculated for unweighted regression and in both cases for weighted regression.

#### 4.2.3. Simulation Studies

In the Monte Carlo simulation studies comparing the “phenotypic” and “genotypic” (unweighted and two ways weighted) estimates of the additive–by–additive–by–additive interaction of QTL effects the following variants of the selected parameters were adopted. The true value of the parameter connected with additive × additive × additive interaction effect was assumed to be equal to 15 (*aaa* = 15) and the total mean value of the quantitative trait to 100. In all 500 homozygous lines were analyzed. Each homozygous line was represented by ten plants. We assumed genotyping for 1000 molecular markers distributed in ten chromosomes of 100 cM. The number of molecular markers per chromosome was equal to 100. The distance between markers was equal to 1 cM. The number of QTLs affecting the trait was 14 (each with an additive effect of two). A QTL was marked perfectly so that the marker and QTLs were in complete linkage disequilibrium with no recombination. We allocated one QTL in chromosome 1 (Q1), two QTLs in chromosomes 3 (Q2 and Q3), 5 (Q4 and Q5), and three QTLs in chromosomes 7 (Q6, Q7 and Q8), 9 (Q9, Q10 and Q11), and 10 (Q12, Q13 and Q14) (Table 1). We assumed the action of five epistatic pairs: Q1–Q2, Q4–Q5, Q6–Q7, Q8–Q11, and Q1–Q14 (Figure 1). Thus, we defined two epistatic pairs in the same chromosome: Q4 and Q5 (in chromosome 5) and Q6 and Q7 (in chromosome 7), and three epistatic pairs in distinct chromosomes. Equal epistatic effects were assumed for all five pairs in two variants: two or zero (no epistasis). The number of QTL–QTL–QTL threes with additive–by–additive–by–additive interaction effects that affected the trait was assumed to be one, two, three or five. The QTLs with additive–by–additive–by–additive interaction effects were located in one chromosome (three triples on one chromosome and two triples on two chromosomes), two chromosomes or three chromosomes (each QTL was in a different chromosome) (Table 1). Effects of particular triples of genes were assumed to be: (i) equal for all triples, or (ii) one QTL–QTL–QTL triple effect was much large than the other (for two triples: 10 and 5; for three triples: 9, 3 and 3; for five triples: 7, 2, 2, 2 and 2). The error variance was equal to 5 of 10. A total of 10,000 data sets containing the vector of phenotypic observations and vectors of marker genotype observations were generated for each combination of the parameters. For each data set the line variances were estimated and the additive–by–additive–by–additive interaction effect was estimated by the phenotypic method $\left({\hat{aaa}}_{jp}\right)$, and by unweighted $\left({\hat{aaa}}_{jgu}\right)$ and weighted $\left({\hat{aaa}}_{jgw1},\;{\hat{aaa}}_{jgw2}\right)$ $\left(j=1,\;2,\;\ldots ,\;10,000\right)$ versions of the genotypic method presented above. Additionally, the coefficients of determination $R_{j}^{2}$ were estimated. Then, the mean values of parameter estimates ${\bar{\hat{aaa}}}_{p}$, ${\bar{\hat{aaa}}}_{gu}$, ${\bar{\hat{aaa}}}_{gw1}$ and ${\bar{\hat{aaa}}}_{gw2}$ for each series were calculated, together with the mean squared errors. Mean values of *R*^2^ were calculated. Statistical analysis was performed using GenStat 22nd edition [77].

## 5. Conclusions

In practical research, one should expect estimates of the total effect of additive–by–additive–by–additive for triplets of gene–by–gene–by–gene interactions quantitative traits to be smaller than the total phenotype estimate. If a different situation is found, it should be carefully analyzed, checking whether the genetic assumptions of correct segregation and lack of close linkage between markers are met. Weighted multiple linear regression is useful for estimating the parameter associated with additive–by–additive–by–additive of the gene–by–gene–by–gene interaction effect. Regardless of the variant of the approach in the weighted regression, an improvement in the estimation of the parameter under consideration was obtained compared to that obtained using unweighted regression. This indicates the applicability of this estimation method in various experimental situations. The results indicate that the estimation of additive–by–additive–by–additive interaction effects by the weighted regression method can be applied to different plant species.

## Figures and Tables

**Figure 1 ijms-24-10043-f001:**
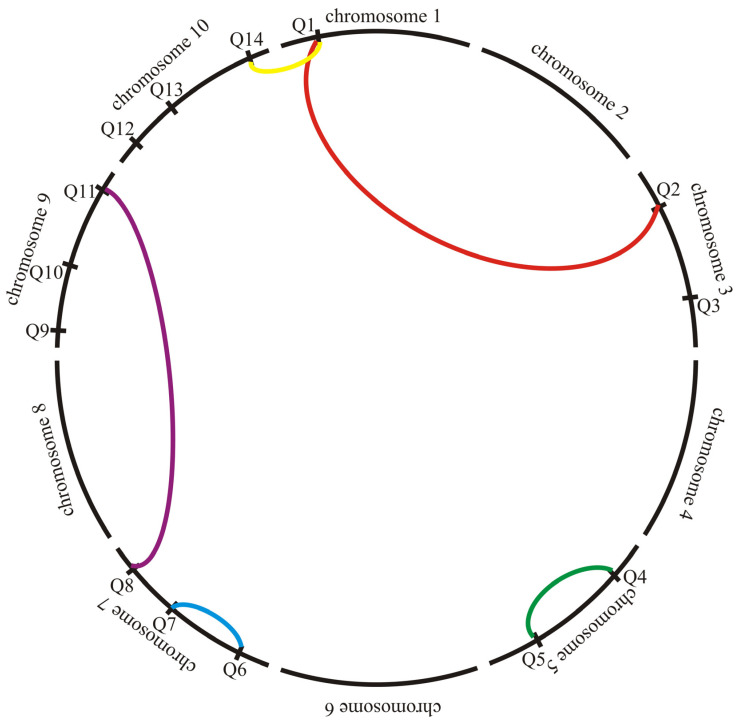
Localization on chromosomes of assumed 14 QTLs and five epistatic pairs.

**Figure 2 ijms-24-10043-f002:**
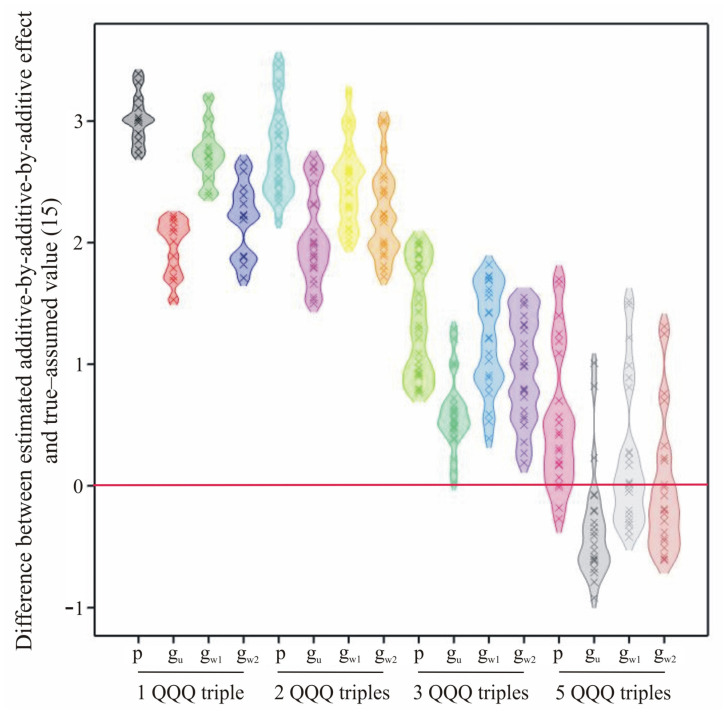
Differences between estimated additive–by–additive–by–additive effect and true–assumed value (15) for particular methods of estimation and different number of QTL–QTL–QTL triples. p—phenotypic method, g_u_—unweighted regression, g_w1_—weighted regression: weighted triple interaction effects only, g_w2_—weighted regression: weighted all effects: QTL, epistatic and triple interaction.

**Table 1 ijms-24-10043-t001:** The number of QTL–QTL–QTL triples assumed, their designations and locations on the chromosomes, and the coding of individual variants.

Variant Code	The Number of Triples	QTL–QTL–QTL Triples	The Position of Triples on Chromosomes
C01	1	Q6Q7Q8	all on one chromosome
C02	1	Q1Q2Q3	on two different chromosomes
C03	1	Q1Q2Q4	on three different chromosomes
C04	2	Q6Q7Q8, Q9Q10Q11	all on one chromosome
C05	2	Q1Q2Q3, Q4Q5Q6	on two different chromosomes
C06	2	Q1Q2Q4, Q3Q5Q7	on three different chromosomes
C07	3	Q6Q7Q8, Q9Q10Q11, Q12Q13Q14	all on one chromosome
C08	3	Q1Q2Q3, Q4Q5Q6, Q7Q10Q11	on two different chromosomes
C09	3	Q1Q2Q4, Q3Q5Q7, Q4Q6Q9	on three different chromosomes
C10	5	Q6Q7Q8, Q9Q10Q11, Q12Q13Q14, Q1Q2Q3, Q1Q4Q5	three triples on one chromosome and two triples on two chromosomes
C11	5	Q1Q2Q3, Q2Q4Q5, Q4Q6Q7, Q1Q9Q10, Q1Q12Q14	on two different chromosomes
C12	5	Q1Q2Q4, Q3Q5Q6, Q1Q9Q12, Q2Q10Q14, Q5Q10Q12	on three different chromosomes

**Table 2 ijms-24-10043-t002:** Phenotypic and genotypic estimates of the total additive–additive–additive effect, mean squared errors and mean of coefficients of determination
$R_{j}^{2}$ obtained in the simulation study for: the error variance was equal to 5, no epistasis, equal effects for all triples.

Variant Code	Number of QTL–QTL–QTL Interaction Effects	Number of Chromosomes with QTL Triples	Estimate				Mean Squared Error For				*R* ^2^
			${\bar{\hat{aaa}}}_{p}$	${\bar{\hat{aaa}}}_{gu}$	${\bar{\hat{aaa}}}_{gw1}$	${\bar{\hat{aaa}}}_{gw2}$	${\hat{aaa}}_{p}$	${\hat{aaa}}_{gu}$	${\hat{aaa}}_{gw1}$	${\hat{aaa}}_{gw2}$	
C01	1	1	17.99	17.01	17.63	17.22	0.23	0.31	0.25	0.42	0.89
C02	1	2	18.01	17.09	17.71	17.32	0.27	0.33	0.28	0.41	0.87
C03	1	3	18.32	17.12	18.02	17.59	0.28	0.40	0.33	0.44	0.92
C04	2	1	17.55	16.82	17.30	16.91	0.33	0.42	0.38	0.43	0.84
C05	2	2	17.80	17.09	17.62	17.39	0.37	0.44	0.40	0.50	0.88
C06	2	3	18.27	17.62	17.93	17.52	0.37	0.47	0.42	0.55	0.95
C07	3	1	15.97	15.48	15.82	15.55	0.45	0.53	0.49	0.66	0.94
C08	3	2	16.43	15.66	16.21	16.09	0.47	0.59	0.51	0.68	0.87
C09	3	3	16.91	16.01	16.72	16.49	0.61	0.72	0.68	0.73	0.92
C10	5	1, 2	15.57	14.42	15.02	14.77	0.55	0.71	0.60	0.82	0.90
C11	5	2	16.25	14.93	15.99	15.21	0.56	0.78	0.69	0.88	0.87
C12	5	3	16.66	15.82	16.49	16.31	0.62	0.82	0.77	1.01	0.91

**Table 3 ijms-24-10043-t003:** Phenotypic and genotypic estimates of the total additive–additive–additive effect, mean squared errors and mean of coefficients of determination $R_{j}^{2}$ obtained in the simulation study for: the error variance was equal to 5, no epistasis, different effects for triples.

Variant Code	Number of QTL–QTL–QTL Interaction Effects	Number of Chromosomes with QTL Triples	Estimate				Mean Squared Error For				*R* ^2^
			${\bar{\hat{aaa}}}_{p}$	${\bar{\hat{aaa}}}_{gu}$	${\bar{\hat{aaa}}}_{gw1}$	${\bar{\hat{aaa}}}_{gw2}$	${\hat{aaa}}_{p}$	${\hat{aaa}}_{gu}$	${\hat{aaa}}_{gw1}$	${\hat{aaa}}_{gw2}$	
C04	2	1	17.48	16.97	17.42	16.89	0.30	0.44	0.37	0.44	0.84
C05	2	2	17.89	17.32	17.59	17.42	0.41	0.47	0.48	0.52	0.95
C06	2	3	18.33	17.58	17.99	17.99	0.42	0.49	0.45	0.59	0.83
C07	3	1	15.92	15.47	15.89	15.73	0.47	0.51	0.48	0.59	0.90
C08	3	2	16.58	15.60	16.43	15.98	0.51	0.60	0.53	0.66	0.87
C09	3	3	17.02	16.21	16.69	16.55	0.59	0.77	0.63	0.89	0.88
C10	5	1, 2	15.52	14.62	15.17	14.80	0.56	0.73	0.69	0.79	0.88
C11	5	2	16.19	15.23	15.89	15.70	0.59	0.74	0.63	0.90	0.92
C12	5	3	16.70	16.01	16.52	16.25	0.73	1.01	0.88	1.17	0.94

**Table 4 ijms-24-10043-t004:** Phenotypic and genotypic estimates of the total additive–additive–additive effect, mean squared errors and mean of coefficients of determination $R_{j}^{2}$ obtained in the simulation study for: the error variance was equal to 5, presence of epistasis, equal effects for all triples.

**Variant Code**	**Number of** **QTL–QTL–QTL** **Interaction Effects**	**Number of** **Chromosomes with QTL Triples**	**Estimate**				**Mean Squared Error For**				** *R* ** ** ^2^ **
			${\bar{\hat{aaa}}}_{p}$	${\bar{\hat{aaa}}}_{gu}$	${\bar{\hat{aaa}}}_{gw1}$	${\bar{\hat{aaa}}}_{gw2}$	${\hat{aaa}}_{p}$	${\hat{aaa}}_{gu}$	${\hat{aaa}}_{gw1}$	${\hat{aaa}}_{gw2}$	
C01	1	1	17.72	16.53	17.39	16.71	0.19	0.28	0.21	0.39	0.89
C02	1	2	17.84	16.72	17.53	16.82	0.25	0.30	0.27	0.40	0.94
C03	1	3	18.01	16.89	17.72	17.19	0.25	0.37	0.32	0.42	0.90
C04	2	1	17.40	16.66	17.07	16.72	0.30	0.39	0.33	0.40	0.84
C05	2	2	17.71	16.79	17.55	17.01	0.36	0.41	0.40	0.49	0.90
C06	2	3	17.99	16.50	17.62	17.17	0.34	0.40	0.37	0.54	0.86
C07	3	1	15.79	15.03	15.39	15.19	0.40	0.49	0.42	0.58	0.89
C08	3	2	16.29	15.38	16.03	15.57	0.44	0.53	0.49	0.66	0.92
C09	3	3	16.77	15.55	16.59	16.02	0.58	0.69	0.66	0.70	0.89
C10	5	1, 2	15.42	14.29	14.77	14.99	0.50	0.63	0.52	0.77	0.94
C11	5	2	16.09	14.70	15.81	15.33	0.54	0.75	0.62	0.81	0.95
C12	5	3	16.40	14.92	16.22	15.76	0.59	0.81	0.73	0.99	0.88

**Table 5 ijms-24-10043-t005:** Phenotypic and genotypic estimates of the total additive–additive–additive effect, mean squared errors and mean of coefficients of determination $R_{j}^{2}$ obtained in the simulation study for: the error variance was equal to 5, presence of epistasis, dif-ferent effects for triples.

Variant Code	Number of QTL–QTL–QTL Interaction Effects	Number of Chromosomes with QTL Triples	Estimate				Mean Squared Error For				** *R* ** ** ^2^ **
			$${\bar{\hat{aaa}}}_{p}$$	$${\bar{\hat{aaa}}}_{gu}$$	$${\bar{\hat{aaa}}}_{gw1}$$	$${\bar{\hat{aaa}}}_{gw2}$$	$${\hat{aaa}}_{p}$$	$${\hat{aaa}}_{gu}$$	$${\hat{aaa}}_{gw1}$$	$${\hat{aaa}}_{gw2}$$	
C04	2	1	17.19	16.53	16.98	16.77	0.33	0.40	0.37	0.48	0.95
C05	2	2	17.37	16.80	17.12	16.97	0.36	0.44	0.41	0.53	0.83
C06	2	3	17.42	16.99	17.28	17.17	0.36	0.45	0.39	0.61	0.85
C07	3	1	15.77	15.22	15.59	15.36	0.39	0.52	0.42	0.70	0.91
C08	3	2	15.90	15.39	16.22	15.78	0.42	0.59	0.51	0.68	0.95
C09	3	3	16.82	15.66	16.70	16.32	0.55	0.68	0.60	0.73	0.87
C10	5	1, 2	14.99	14.38	14.73	14.55	0.51	0.63	0.53	0.79	0.89
C11	5	2	15.07	14.42	14.80	14.55	0.54	0.77	0.66	0.93	0.90
C12	5	3	15.18	14.66	14.98	14.81	0.58	0.83	0.69	0.97	0.88

**Table 6 ijms-24-10043-t006:** Phenotypic and genotypic estimates of the total additive–additive–additive effect, mean squared errors and mean of coefficients of determination $R_{j}^{2}$ obtained in the simulation study for: the error variance was equal to 10, no epistasis, equal effects for all triples.

**Variant Code**	**Number of** **QTL–QTL–QTL** **Interaction Effects**	**Number of** **Chromosomes with QTL Triples**	**Estimate**				**Mean Squared Error For**				*R^2^*
			${\bar{\hat{aaa}}}_{p}$	${\bar{\hat{aaa}}}_{gu}$	${\bar{\hat{aaa}}}_{gw1}$	${\bar{\hat{aaa}}}_{gw2}$	${\hat{aaa}}_{p}$	${\hat{aaa}}_{gu}$	${\hat{aaa}}_{gw1}$	${\hat{aaa}}_{gw2}$	
C01	1	1	18.03	17.09	17.77	17.39	0.44	0.57	0.49	0.73	0.91
C02	1	2	18.11	17.20	17.79	17.45	0.49	0.60	0.54	0.80	0.87
C03	1	3	18.39	17.22	18.19	17.66	0.54	0.77	0.63	0.91	0.95
C04	2	1	17.61	16.89	17.41	17.01	0.64	0.81	0.75	0.94	0.86
C05	2	2	17.88	17.31	17.77	17.52	0.66	0.79	0.69	0.94	0.87
C06	2	3	18.44	17.63	18.02	17.77	0.68	0.91	0.80	0.99	0.88
C07	3	1	16.02	15.54	15.91	15.62	0.77	0.91	0.84	1.02	0.90
C08	3	2	16.51	15.72	16.43	16.17	0.82	0.93	0.89	1.05	0.92
C09	3	3	16.98	16.29	16.55	16.40	0.83	0.93	0.88	1.11	0.93
C10	5	1, 2	14.99	14.29	14.70	14.42	0.77	0.89	0.81	0.99	0.87
C11	5	2	15.17	14.40	15.01	14.62	0.83	0.99	0.91	1.12	0.84
C12	5	3	15.52	14.80	15.27	14.99	0.90	1.07	0.97	1.23	0.91

**Table 7 ijms-24-10043-t007:** Phenotypic and genotypic estimates of the total additive–additive–additive effect, mean squared errors and mean of coefficients of determination $R_{j}^{2}$ obtained in the simulation study for: the error variance was equal to 10, no epistasis, different effects for triples.

Variant Code	Number of QTL–QTL–QTL Interaction Effects	Number of Chromosomes with QTL Triples	Estimate				Mean Squared Error For				*R^2^*
			${\bar{\hat{aaa}}}_{p}$	${\bar{\hat{aaa}}}_{gu}$	${\bar{\hat{aaa}}}_{gw1}$	${\bar{\hat{aaa}}}_{gw2}$	${\hat{aaa}}_{p}$	${\hat{aaa}}_{gu}$	${\hat{aaa}}_{gw1}$	${\hat{aaa}}_{gw2}$	
C04	2	1	17.51	17.00	17.49	17.23	0.55	0.77	0.61	0.80	0.90
C05	2	2	17.93	17.49	17.60	17.55	0.72	0.91	0.80	0.97	0.87
C06	2	3	18.50	17.69	18.23	18.01	0.79	0.99	0.88	1.07	0.94
C07	3	1	16.09	15.52	15.91	15.79	0.82	0.93	0.87	1.11	0.89
C08	3	2	16.22	15.63	16.09	15.99	0.91	1.21	0.99	1.33	0.86
C09	3	3	16.90	15.99	16.73	16.28	0.92	1.27	1.02	1.44	0.85
C10	5	1, 2	15.29	14.47	15.03	14.71	0.88	0.99	0.92	1.03	0.90
C11	5	2	15.31	14.59	15.28	14.92	0.91	1.08	0.97	1.20	0.87
C12	5	3	15.57	14.66	15.02	15.23	0.97	1.22	1.12	1.40	0.88

**Table 8 ijms-24-10043-t008:** Phenotypic and genotypic estimates of the total additive–additive–additive effect, mean squared errors and mean of coefficients of determination $R_{j}^{2}$ obtained in the simulation study for: the error variance was equal to 10, presence of epistasis, equal effects for all triples.

Variant Code	Number of QTL–QTL–QTL Interaction Effects	Number of Chromosomes with QTL Triples	Estimate				Mean Squared Error For				*R* ^2^
			${\bar{\hat{aaa}}}_{p}$	${\bar{\hat{aaa}}}_{gu}$	${\bar{\hat{aaa}}}_{gw1}$	${\bar{\hat{aaa}}}_{gw2}$	${\hat{aaa}}_{p}$	${\hat{aaa}}_{gu}$	${\hat{aaa}}_{gw1}$	${\hat{aaa}}_{gw2}$	
C01	1	1	17.77	16.69	17.42	16.88	0.30	0.41	0.37	0.73	0.89
C02	1	2	17.90	16.79	17.66	16.89	0.42	0.55	0.49	0.79	0.94
C03	1	3	18.19	17.17	17.89	17.23	0.44	0.71	0.59	0.79	0.95
C04	2	1	17.42	16.70	17.11	16.99	0.51	0.77	0.57	0.82	0.88
C05	2	2	17.69	16.88	17.58	17.20	0.52	0.71	0.60	0.80	0.97
C06	2	3	18.08	16.91	17.71	17.44	0.58	0.77	0.66	0.84	0.86
C07	3	1	15.80	15.12	15.53	15.27	0.69	0.80	0.77	0.91	0.90
C08	3	2	16.32	15.44	16.42	16.33	0.73	0.92	0.81	0.99	0.92
C09	3	3	16.83	15.61	16.62	16.51	0.74	0.98	0.89	1.02	0.87
C10	5	1, 2	15.21	14.08	14.63	14.49	0.88	1.00	0.93	1.17	0.86
C11	5	2	15.44	14.32	14.77	14.57	0.92	1.19	1.04	1.31	0.91
C12	5	3	15.70	14.79	15.22	15.01	0.98	1.33	1.07	1.40	0.95

**Table 9 ijms-24-10043-t009:** Phenotypic and genotypic estimates of the total additive–additive–additive effect, mean squared errors and mean of coefficients of determination $R_{j}^{2}$ obtained in the simulation study for: the error variance was equal to 10, presence of epistasis, different effects for triples.

Variant Code	Number of QTL–QTL–QTL Interaction Effects	Number of Chromosomes with QTL Triples	Estimate				Mean Squared Error For				*R* ^2^
			${\bar{\hat{aaa}}}_{p}$	${\bar{\hat{aaa}}}_{gu}$	${\bar{\hat{aaa}}}_{gw1}$	${\bar{\hat{aaa}}}_{gw2}$	${\hat{aaa}}_{p}$	${\hat{aaa}}_{gu}$	${\hat{aaa}}_{gw1}$	${\hat{aaa}}_{gw2}$	
C04	2	1	17.33	16.55	17.02	16.81	0.49	0.73	0.52	0.80	0.94
C05	2	2	17.49	17.01	17.19	16.99	0.55	0.80	0.58	0.92	0.91
C06	2	3	17.66	17.09	17.31	17.24	0.61	0.92	0.77	0.99	0.87
C07	3	1	15.80	15.49	15.76	15.50	0.66	0.88	0.79	0.92	0.85
C08	3	2	15.93	15.52	15.91	15.80	0.70	0.93	0.82	1.02	0.88
C09	3	3	16.99	15.72	16.82	16.55	0.82	1.09	0.97	1.25	0.90
C10	5	1, 2	14.73	14.21	14.58	14.39	0.79	0.98	0.85	1.07	0.95
C11	5	2	14.82	14.39	14.68	14.40	0.85	1.12	0.93	1.30	0.91
C12	5	3	15.01	14.52	14.95	14.77	0.98	1.21	1.07	1.44	0.92

## Data Availability

The data presented in this study are available on request from the corresponding author.

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
