# Peer review of "A Comparison of Methods to Estimate Additive–by–Additive–by–Additive of QTL×QTL×QTL Interaction Effects by Monte Carlo Simulation Studies"

_ijms, 2023, doi:10.3390/ijms241210043_

Round 1

Reviewer 1 Report

The Authors undertook the very difficult task of assessing the triple interaction of genes determining a quantitative trait. To date, the literature has dealt with the estimation of gene effects (QTLs) and/or their epistatic effects (QTL-by-QTL), but there are no publications considering QTL-by-QTL-by-QTL interaction. The Authors' very novel approach should be emphasized.

The second very important aspect is how to approach the issue. The authors used simulation studies. Based on 84 different sets of parameter combinations assumed in the simulation studies, the authors exhaust, in my opinion, the spectrum of cases found in real-world experiments. One could assume still other experimental situations (different error variance, more triples), however, in my opinion, the situations presented are sufficient.

The Authors compare two main estimation methods: based on phenotypic observations only and based on marker observations. While in the latter case the number of QTLs is very easy to estimate, in the case of the phenotypic method I missed providing a formula for the number of effective factors. I would recommend supplementing the manuscript with this formula.

Paper needs minor revision.

Author Response

Response to Reviewer 1 Comments

Reviewer #1

Point 1: The Authors undertook the very difficult task of assessing the triple interaction of genes determining a quantitative trait. To date, the literature has dealt with the estimation of gene effects (QTLs) and/or their epistatic effects (QTL-by-QTL), but there are no publications considering QTL-by-QTL-by-QTL interaction. The Authors' very novel approach should be emphasized.

Response: Thank you very much for appreciating the results of our research.

Point 2: The second very important aspect is how to approach the issue. The authors used simulation studies. Based on 84 different sets of parameter combinations assumed in the simulation studies, the authors exhaust, in my opinion, the spectrum of cases found in real-world experiments. One could assume still other experimental situations (different error variance, more triples), however, in my opinion, the situations presented are sufficient.

Response: Thank you very much.

Point 3: The Authors compare two main estimation methods: based on phenotypic observations only and based on marker observations. While in the latter case the number of QTLs is very easy to estimate, in the case of the phenotypic method I missed providing a formula for the number of effective factors. I would recommend supplementing the manuscript with this formula.

Response: The manuscript was supplemented with a formula for the number of effective factors (the number of genes calculated based on phenotypic observations only).

Reviewer 2 Report

The study aims to compare two methods of estimating quantitative effect of additive-by-additive-by-additive interactions. Authors conduct Monte Carlo simulation studies with 84 different experimental situations. The findings indicate that weighted regression is the preferred method for estimating additive-by-additive-by-additive interaction effects in QTL-QTL-QTL triples. The authors argue that weighted regression provides more accurate results closer to the true values of total interaction effects compared to unweighted regression. The determination coefficients of the proposed models support the effectiveness of weighted regression in explaining the observed variation in the data. It offers a comprehensive approach by considering diverse experimental situations. The preference for weighted regression has practical implications for breeding programs seeking to select optimal breeding material.

Strengths of the study:

  1. Comprehensive approach: The study incorporates Monte Carlo simulation studies with a wide range of parameter combinations, representing 84 different experimental situations. This comprehensive approach strengthens the reliability and generalizability of the findings.

  2. Novelty: The paper addresses a gap in the literature by exploring and comparing methods for estimating additive-by-additive-by-additive interaction effects in QTL analysis. By highlighting the absence of publications in the open literature on this topic, the study offers a comparison between the two methods to facilitate selection of breeding material to improve agronomic traits of crops.

  3. Practical implications: The preference for weighted regression as a method for estimating additive-by-additive-by-additive interaction effects has practical implications for breeding programs. By providing results closer to the true values of total interaction effects, the study offers a valuable tool for decision-making in selecting plant breeding material.

Weakness of the study:

  1. Lack of empirical validation: While the study employs Monte Carlo simulation studies, which the authors highlight as a reliable tool for such simulations, it is essential to note that these simulations may not fully capture the complexities and nuances of real-world genetic interactions. The findings should be interpreted with caution until further empirical validation is conducted.

  2. Lack of discussion on empirical findings: The authors don’t discuss the comparison presented in this study in context of their previous findings. The two methods presented here builds on their previous work, and it would be beneficial for the readers to understand how these comparison fair to their previous work. It would be also great to highlight how their work harmonizes with empirical findings of various QTL/GWA studies of complex traits, that they highlight in the introduction.

Some specific points:

Alternative explanation of lower estimate of genotypic effect can be due to unaccounted pleiotropic effect from other loci (more than 3), which the authors allude to in line 54-55 or as elegantly described by Liu et. al https://doi.org/10.1016/j.cell.2019.04.014. This is reflected in the results, where the phenotypic values are closer to the true positive or five QTL–QTL–QTL triples, and is probably being overestimated for one triple (discussed in line 268-269). The findings of the study indeed harmonize with the empirical findings, where small additive effect of multiple genes affect a trait, and it would be nice to see in a figure how as the number of QTL interactions increase, the trait value converges to the true positive. I believe this result adds more value to the study.

The results on co-efficients of determination is lacking, and the take home message of the metric is not clear.

It would be good to cite references for the selection of the parameters chosen for comparisons and contextualize them against empirical estimates (line 257).

The clarity of the paper is good, however, some minor grammatical errors can be improved.

Author Response

Response to Reviewer 2 Comments

Reviewer #2

Point 1: The study aims to compare two methods of estimating quantitative effect of additive-by-additive-by-additive interactions. Authors conduct Monte Carlo simulation studies with 84 different experimental situations. The findings indicate that weighted regression is the preferred method for estimating additive-by-additive-by-additive interaction effects in QTL-QTL-QTL triples. The authors argue that weighted regression provides more accurate results closer to the true values of total interaction effects compared to unweighted regression. The determination coefficients of the proposed models support the effectiveness of weighted regression in explaining the observed variation in the data. It offers a comprehensive approach by considering diverse experimental situations. The preference for weighted regression has practical implications for breeding programs seeking to select optimal breeding material.

Response: Thank you very much for appreciating the results of our research.

Strengths of the study:

Point 2: Comprehensive approach: The study incorporates Monte Carlo simulation studies with a wide range of parameter combinations, representing 84 different experimental situations. This comprehensive approach strengthens the reliability and generalizability of the findings.

Response: Thank you very much.

Point 3: Novelty: The paper addresses a gap in the literature by exploring and comparing methods for estimating additive-by-additive-by-additive interaction effects in QTL analysis. By highlighting the absence of publications in the open literature on this topic, the study offers a comparison between the two methods to facilitate selection of breeding material to improve agronomic traits of crops.

Response: Thank you very much.

Point 4: Practical implications: The preference for weighted regression as a method for estimating additive-by-additive-by-additive interaction effects has practical implications for breeding programs. By providing results closer to the true values of total interaction effects, the study offers a valuable tool for decision-making in selecting plant breeding material.

Response: Thank you very much.

Weakness of the study:

Point 5: Lack of empirical validation: While the study employs Monte Carlo simulation studies, which the authors highlight as a reliable tool for such simulations, it is essential to note that these simulations may not fully capture the complexities and nuances of real-world genetic interactions. The findings should be interpreted with caution until further empirical validation is conducted.

Response: We agree with the Reviewer that the simulation studies conducted may not fully capture the complexities and nuances of real–world genetic interactions. We also agree that the results obtained should be interpreted with caution until further empirical validation. We have revised the manuscript emphasizing these aspects.

Point 6: Lack of discussion on empirical findings: The authors don’t discuss the comparison presented in this study in context of their previous findings. The two methods presented here builds on their previous work, and it would be beneficial for the readers to understand how these comparison fair to their previous work. It would be also great to highlight how their work harmonizes with empirical findings of various QTL/GWA studies of complex traits, that they highlight in the introduction.

Response: There are only three publications in the literature with results on triple interaction evaluation. These are the results of our previous studies. We have completed the manuscript and refer to all three of these articles in the discussion. We have also added information about the applicability of the proposed methods for estimating the additive-additive effect of QTL-QTL-QTL triple interaction for association mapping.

Some specific points:

Point 7: Alternative explanation of lower estimate of genotypic effect can be due to unaccounted pleiotropic effect from other loci (more than 3), which the authors allude to in line 54-55 or as elegantly described by Liu et. al https://doi.org/10.1016/j.cell.2019.04.014. This is reflected in the results, where the phenotypic values are closer to the true positive or five QTL–QTL–QTL triples, and is probably being overestimated for one triple (discussed in line 268-269). The findings of the study indeed harmonize with the empirical findings, where small additive effect of multiple genes affect a trait, and it would be nice to see in a figure how as the number of QTL interactions increase, the trait value converges to the true positive. I believe this result adds more value to the study.

Response: The pleiotropic effect may affect the evaluation of the interaction effect. However, in this paper we consider a single quantitative trait and the problem of considering/not considering pleiotropic effect does not apply in these cases. Thank you very much for pointing out a very interesting article by Liu et al. (https://doi.org/10.1016/j.cell.2019.04.014), which we take the liberty to reference in our manuscript. Thank you, for appreciating our study and noting that the results obtained indeed harmonize with the empirical results, and for suggesting that the manuscript be supplemented with a figure showing how, as the number of QTL interactions increases, the value of the trait converges to the true assumed value. We have added such a figure.

Point 8: The results on co-efficients of determination is lacking, and the take home message of the metric is not clear.

Response: Coefficients of determination are used to measure what percentage of the explanatory variables explain the variation in the dependent variable, which in our case is the observed quantitative trait. Assuming that the model includes QTLs with additive effects, pairs of QTLs with epistatic effects, and triple QTLs with additive by additive by additive effects, then R2 is a measure of the fit of the proposed model. We have supplemented the Results section with a detailed description regarding coefficients of determination.

Point 9: It would be good to cite references for the selection of the parameters chosen for comparisons and contextualize them against empirical estimates (line 257).

Response: Thank you very much for your valuable comment. We have supplemented the manuscript with the following text: "In simulation studies, the number of lines analyzed is most often assumed to be as high as 500 or 2000 [45,46]. Viana et al. [46] analyzed 400 plants, genotyped for 975 SNPs distributed in 10 chromosomes of 100 cM. One thousand SNP markers were considered in their study by Crawford et al. [47]. The ten chromosomes include: Anthoxanthum odoratum, Brassica campestris, Sorghum sudanense, Zea maize. Practical experience shows that the number of genes located on individual chromosomes varies greatly [48–51]. The number of QTLs assumed in the present study was 14. The same number of QTLs was presented in their studies by Wu [52] and Balestre et al. [53]. The assumed number of five epistatic pairs was a result observed in many practical experiments [54–57]. Simulation studies based on 10000 generations were also presented by Goutelle et al. [58], Avery et al. [59], Johnson et al. [60], Sorojsrisom et al. [61], and Wang et al. [62], among others."
